# Widely Targeted Metabolomic Profiling Combined with Transcriptome Analysis Provides New Insights into Lipid Biosynthesis in Seed Kernels of *Pinus koraiensis*

**DOI:** 10.3390/ijms241612887

**Published:** 2023-08-17

**Authors:** Yan Li, Yujin Xu, Rui Han, Lin Liu, Xiaona Pei, Xiyang Zhao

**Affiliations:** 1Jilin Provincial Key Laboratory of Tree and Grass Genetics and Breeding, College of Forestry and Grassland Science, Jilin Agricultural University, Changchun 130118, China; ly2019nefu@163.com (Y.L.); xuyjphd@163.com (Y.X.); hanrui4579@jlau.edu.cn (R.H.); liu1601622042@163.com (L.L.); 2College of Life Science, Jilin Agricultural University, Changchun 130118, China; 3College of Horticulture, Jilin Agricultural University, Changchun 130118, China

**Keywords:** *P. koraiensis*, lipid biosynthesis, differentially expressed metabolites, differentially expressed genes, WGCNA

## Abstract

Lipid-rich *Pinus koraiensis* seed kernels are highly regarded for their nutritional and health benefits. To ascertain the molecular mechanism of lipid synthesis, we conducted widely targeted metabolomic profiling together with a transcriptome analysis of the kernels in *P. koraiensis* cones at various developmental stages. The findings reveal that 148 different types of lipid metabolites, or 29.6% of total metabolites, are present in kernels. Among those metabolites, the concentrations of linoleic acid, palmitic acid, and α-linolenic acid were higher, and they steadily rose as the kernels developed. An additional 10 hub genes implicated in kernel lipid synthesis were discovered using weighted gene co-expression network analysis (WGCNA), gene interaction network analysis, oil body biosynthesis, and transcriptome analysis. This study used lipid metabolome and transcriptome analyses to investigate the mechanisms of key regulatory genes and lipid synthesis molecules during kernel development, which served as a solid foundation for future research on lipid metabolism and the creation of *P. koraiensis* kernel food.

## 1. Introduction

*P. koraiensis*, sometimes referred to as nut pine, is a perennial evergreen conifer species that is mostly found in northeast China, Japan, the Korean Peninsula, and the Far East area of Russia. It is a member of the Pinaceae family. It is an essential species of oil and wood tree with great ecological and monetary importance [1,2,3]. It is widely used in construction, transportation, furniture, and other fields because of its soft wood material, simplicity of industrial processing, resistance to corrosion, and lack of tendency to deform after processing. This tree species is also thought to be the best choice for producing high-quality large-diameter timber [4,5,6]. *P. koraiensis* has monoecious reproductive biological traits, and its fruiting phase lasts for two years. Flowering occurs after June of the first year of pollination, and the cone matures in September of the second year after pollination. The seeds are obovate triangles and the cones are fashioned similar to an oval [7,8]. *P. koraiensis* has very high commercial value and its seeds may be utilized to make tasty nut snacks as well as pine seed oil [9,10]. Pine seeds include high quantities of unsaturated fatty acid minerals, proteins, and other components, which have the advantages of decreasing blood pressure and cholesterol levels, delaying the onset of aging, and greatly enhancing the maintenance of human health [11,12,13]. The extraction of phenols, terpenoids, flavonoids, and fatty acids is at present the primary emphasis of *P. koraiensis* seed research [14,15]. According to the reports, mature *P. koraiensis* seeds contain more than 60% lipids, of which up to 90% are unsaturated fatty acids, primarily linolenic acid, linoleic acid, palmitoleic acid, and eicosapentaenoic acid, which can significantly lower blood lipids and have anti-aging and cosmetic effects [16,17,18,19].

Lipids are one of the main sources of energy storage in plant fruits and the most energy-storing component of numerous nutrients [20,21]. Oil is a mixture of liquid fat and mineral hydrocarbons in plants. Its manufacture entails a series of incredibly complicated physiological and biochemical processes involving several enzyme interactions, and it is also one of the most important metabolic routes in plants [22,23,24]. Intriguingly, lipid substances in plants can be referred to as oils, which are mostly composed of fatty acids and glycerol to produce higher fatty acid glyceride compounds and are stored in the cytoplasm of plants as triacylglycerol (TAG) to act as energy reserves [25,26,27,28,29,30]. Lipids have been consumed increasingly used as a main food, and serve as the primary cause of human aliments [31,32,33]. To study the biological pathways of lipid accumulation, it is necessary to understand the molecular mechanisms affecting lipid synthesis and identify the crucial genes regulating metabolic pathways. The recent explosive growth of multi-omics, including transcriptomics and metabolomics, has established a strong molecular foundation for lipid biosynthesis pathways [34,35]. *Brassica napus* [36], *Glycine max* [37], *Arachis hypofaea* L. [38], *Elaeis guineensis* [39], and *Sesamum indicum* L. [40] were the main subjects of earlier molecular investigations on lipid biosynthesis. For instance, in their analysis of the transcriptome of *B. napus* during development using RNA-Seq technology, Deng et al. discovered 55 differentially expressed genes that encoded enzymes involved in fatty acid production, and the results also revealed that hormones and transcription factors were essential for the production of fatty acids during seed development. Furthermore, the researchers discovered 113 key genes involved in lipid synthesis during the growth of *G. max* L using the same method. These genes are highly expressed throughout seed development, and by controlling the fatty acid biosynthesis pathways, they have an impact on lipid synthesis during seed development.

The most crucial step in the growth of a plant is seed development, which also shows the effects of genetic inheritance [41]. The reproductive organs of gymnosperms are macroconidia and microstrobilus, which are distinct from those of angiosperms. They are often globular or spike-shaped and affixed to the tips of leaves or branches. The primary source of energy for plants to transition from vegetative growth to reproductive growth are the reserves stored in branches and needles [42,43]. Enzymes [44], hormones [45], and gene expression [46] are other important elements in the development of seeds in addition to environmental factors. For instance, the tomato (*Solanum lycopersicum*) enzymes choline (COD) and betaine aldehyde dehydrogenase (BADH) play a key role in the production of glycinebetaine, which expedites the development of tomato fruits [47]. Additionally, polyamines can control strawberry (*Fragaria ananassa*) fruit development by balancing the levels of ethylene, auxin, and abscisic acid [48].

Despite the high lipid content of *P. koraiensis* seeds, little is known about the types and amounts of lipids they contain or about the genes involved in lipid production. The interaction between related metabolites and genes during lipid creation is determined in this work by combining extensive targeted metabolomics and transcriptomics studies to investigate the biosynthesis process of lipids in various stages of *P. koraiensis* seed kernel development. The findings offer a theoretical foundation for the commercialization and use of *P. koraiensis* nuts.

## 2. Results

### 2.1. Observation and Characterization of Cone and Seed Development

*P. koraiensis* cones typically develop in the second year after the first pollination. From the second year of cone growth until the cones were fully mature, samples were taken. We collected samples at 130, 160, 190, 220, and 240 days of the year as test materials and labeled them as S1–S5, according to the variations in the development time. Analyzing the phenotypic data revealed that cone length, cone width, seed length, seed width, cone weight, and seed weight all increased gradually throughout the course of development, with S1–S3 showing the fastest growth (Figure 1).

### 2.2. Lipid Metabolite Profile for Seed Kernel Development

The seed kernel was separated from the seed at five stages for highly focused metabolomic profiling and transcriptome analysis to investigate the link between metabolites connected to lipid synthesis and genes throughout the seed development process (Appendix A). A total of 148 lipids were discovered in the metabolomics data of lipids in various phases of seed development, 64 of which were free fatty acids, using UPLC–MS/MS (Appendix A). The PCA results demonstrate that lipid metabolite compounds accumulate differentially at several developmental stages. For instance, the levels of lipid metabolite compounds found in stages S1–S2 were considerably different from those found in subsequent stages; although, there were trends in the levels of metabolites at stages S3, S4, and S5. Principal component 1 (PC1) and 2 (PC2) fractions contributed to 19.16% and 60.66% of the overall principal components of the samples, respectively (Figure 2a). The strong correlation between the biological replicates of samples and the notable variations in lipid properties at various developmental stages were both represented in the intragroup correlation (Figure 2b). For additional analysis, Figure 2c depicts the radar map of the lipid content of the 10 most abundant free fatty acids. Palmitaldehyde was the most abundant, followed by α-linolenic acid, 3-hydroxyoctadecanoic acid, octadeca-11E,13E,15Z-trienoic acid, linoleic acid, elaidic acid, punicic acid (9Z,11E,13Z-octadecatrienoic acid), myristic acid, 9,10,13-trihydroxy-11-octadecenoic acid, and 11-octadecanoic acid (Vaccenic acid). From 12.96% in S1 to 78.34% in S5, the 10 free fatty acid content levels as a percentage of total lipids content increased significantly (Figure 2d).

### 2.3. Statistical Analysis of Transcriptome Data and Gene Functional Annotation

To evaluate the potential molecular underpinnings at various kernel developmental stages, global transcriptomic analyses were performed using RNA-seq. The construction of fifteen cDNA libraries with three biological replicates for each stage resulted in the collection of 807,886,606 raw reads. Fastp software was used to filter and verify sequence errors in the raw data, resulting in the final acquisition of 778,549,924 clean reads, ensuring the excellent quality of the raw reads. Additionally, the clean reads’ average Q20, Q30, and GC contents were 96.44%, 91.58%, and 45.28%, respectively (Appendix A). Trinity software [49] was used to splice clean reads and produce a full transcript for a subsequent bioinformatics analysis because no reference genome for this species exists. Different developmental phases were shown by PCA to have substantial changes in gene expression (Figure 3a). A total of 88342 unigenes were annotated, including 74,722 with GO annotations, 61021 with KEGG terms, 87333 with NR database terms, as well as 47468 with KOG annotations, and 62235 with SwissProt database terms. For the annotations obtained, 52274 were common unigenes among all five databases (Figure 3b). In the KOG annotation, 47468 unigenes were further divided into 25 functional groups. The general function prediction-only group (15062 unigenes) had the greatest number of annotated genes, and the cell motility group had the lowest number of annotated genes (16 unigenes). Notably, 1903 unigenes were annotated to lipid transport and metabolism (Figure 3c). Then, 74722 unigenes, comprising 59 level-two functional categorization words, were categorized into biological process, cellular component, and molecular function categories (Figure 3d).

### 2.4. Differentially Expressed Gene Enrichment Analysis

To further identify the differentially expressed genes (DEGs), the identified common DEGs in all groups were subjected to GO and KEGG enrichment analysis, and the results are presented in a circle diagram. There were more DEGs that were enriched in the molecular function and biological process categories than in the cellular component category, according to the top 20 terms identified by the GO functional enrichment analysis (Figure 4a). To clarify the biological functions of DEGs, further enrichment studies for molecular function, biological process, and cellular component terms were also conducted several times (Appendix A). The results indicate that the beta-glucan metabolic process (GO:0051273), glucan biosynthetic process (GO:0009250), and regulation of organelle organization (GO:0033043) are the most significantly enriched biological processes. The most significantly enriched cellular components included an anchored component of the plasma membrane (GO:0046658), DNA packaging complex (GO:0044815), and protein–DNA complex (GO:0032993), while protein heterodimerization activity (GO:0046982), tubulin binding (GO:0015631), and microtubule binding (GO:0008017) were significantly enriched terms in the molecular function category. In addition, the top 20 significantly enriched terms by KEGG analysis demonstrated that the DEGs were principally abundant in plant hormone signal transduction (ko04075), ubiquitin-mediated proteolysis (ko04120), and flavonoid biosynthesis (ko00943), while 63 and 49 DEGs were significantly enriched in fatty acid elongation (ko00062) and the biosynthesis of unsaturated fatty acids (ko01040), respectively (Figure 4b).

### 2.5. Identification and Expression Patterns of Potential Genes Related to Oil Body Biosynthesis in Kernels

Fatty acid de novo synthesis and modification, triacylglycerol (TAG) assembly, and oil body formation make up the majority of plant lipid formation. In this study, the DEGs involved in the lipid biosynthesis synthesis pathway in *P. koraiensis* seed kernels were identified and analyzed. A total of 286 genes were found, including 129 genes for fatty acid biosynthesis in plastids, 149 genes for TAG assembly in the endoplasmic reticulum, and 8 genes for oil body formation. The results of the DEG analyses shows that the biotin carboxylase subunit of heteromeric acetyl-CoA carboxylase (ACC (BC)), ketoacyl-ACP synthase I (KAS *I*), acyl-carrier protein (ACP), and long-chain acyl-CoA synthetase (LACS) are mainly expressed in the S1–S3 stages, suggesting that these genes may be crucial for fatty acid synthesis in the early stage of seed kernel development. The expression levels of the biotin carboxyl carrier protein of heteromeric acetyl-CoA carboxylase (ACC (BCCP)), stearoyl-ACP reductase (SAD), and pyruvate dehydrogenase (PHD) were higher at the end of development, indicating that these genes may be essential for fatty acid synthesis at the end of seed kernel development. In addition, many DEGs (such as glycerol-3-phosphate acyltransferase (GPAT), lysophosphatidic acid acyltransferase (LPAT), and phospholipid:diacylglycerol acyltransferase (PDAT)) were engaged in the assembly process of TAG, demonstrating that these genes had crucial regulatory functions in this process. Moreover, 5 caleosin, 2 oleosin, and 1 steroleosin (STERO) coding genes were found in this study. It is important to note that the expression trends of these 8 genes were similar, all of which were highly expressed at the S5 stage (Figure 5).

### 2.6. WGCNA

Transcriptome data were used to perform weighted gene co-expression network analysis (WGCNA), and a total of 20 co-expressed gene modules were finally identified (Figure 6a). These modules were further associated with cone length, cone width, seed length, seed width, cone weight, seed weight, α-linolenic acid, linoleic acid, and γ-linolenic acid. As shown in Figure 6b, the gene expression levels in the pink, brown, magenta, and black modules are significantly positively correlated with most indices. In addition, there are relatively high correlation coefficients in the pink module, which are 0.63 (*p* = 0.012), 0.44 (*p* = 0.1), −0.66 (*p* = 0.0074), 0.87 (*p* = 2.5 × 10^−0.5^), 0.79 (*p* = 0.00046), 0.87 (*p* = 2.5 × 10^−0.5^), 0.88 (*p* = 1.5 × 10^−0.5^), 0.87 (*p* = 2.5 × 10^−0.5^), and 0.81 (*p* = 0.00025), indicating that the genes in the pink module are crucial for lipid synthesis and seed kernel development.

### 2.7. Gene Interaction Network Analysis

To obtain the hub genes in the pink module, the top 500 genes with kWithin values in the modules were analyzed in a gene interaction network. According to the visualization and screening, 10 hub genes were identified and are shown in Figure 7a. These 10 hub genes were mainly involved in biosynthesis processes, such as alpha-amylase, glycogen phosphorylase, and glucose-1-phosphate adenylyltransferase. The gene expression heatmap is shown in Figure 7b. All 10 hub genes displayed high expressions in the S3–S5 stages.

### 2.8. RT-qPCR Validation of Differentially Expressed Genes

To verify the accuracy of transcriptome sequencing, 9 genes were selected from the obtained DEGs, and their expression levels were verified by RT-qPCR at five different developmental stages of the seed kernel (Appendix A). It was found that the expression patterns of most of the differentially expressed genes identified by RT-qPCR and RNA-seq were consistent, indicating the accuracy and dependability of the transcriptome sequencing data used in this work.

## 3. Discussion

*P. koraiensis* is favored by forest growers and consumers because of its high economic and ecological benefits. The high lipid content of pine nuts accounts for their anti-lipid and anti-hypertension effects [7]. Studies have found that plant lipid metabolic synthesis is an extremely complex process [50]. In this study, the molecular mechanism of lipid biosynthesis in the different developmental stages of *P. koraiensis* seed kernels was investigated by extensive targeted metabolomic profiling and transcriptome analysis. The results of this study can provide a reference for the control network of lipid synthesis in conifers, lipid accumulation metabolites, and molecular mechanisms during seed development.

In organisms, various life activities of tissues and organs are coordinated by many genes, amino acids, proteins, and small-molecule metabolites, involving signal transduction, material synthesis and metabolism, energy transfer, and other important life phenomena [51,52,53]. Metabolomics is a key bridge connecting gene level and phenotypic characteristics and is a science used to study the dynamic changes in endogenous metabolites in organisms after gene variation or environmental changes [54]. Its core function is to quantify metabolites, such as nucleosides, terpenoids, alkaloids, amino acids, and carbohydrates; analyze differential metabolites at various growth and development stages; characterize the changes in metabolites; and provide a critical material basis and genetic information for plant phenotypic variation [55,56,57]. A total of 500 metabolites, including phenolic acids, lipids, amino acids and their derivatives, organic acids, nucleotides and their derivatives, and some other metabolites, were identified by the metabolome of kernels from five developmental stages of *P. koraiensis* seeds. Among them, lipids (29.6%) and amino acids and their derivatives (21.4%) were the most abundant, and the number of their metabolites was more than 100, indicating that these metabolites were the main members of the metabolic profile during the development of *P. koraiensis* seed kernels. In addition, the contents of palmitic acid, linolenic acid, linoleic acid, elaidic acid, and myristic acid were relatively high and were mainly distributed in the later stage. The proportion of the 10 most abundant free fatty acids in the S5 period was as high as 78.34%, indicating that, as the fruit became larger, the contents of lipid metabolites, such as linolenic and linoleic acids, in the kernel also increased, which was similar to the results of a study of *Juglans mandshurica* [58]. In their study, there were abundant lipids in *J. mandshurica*, mainly including linoleic acid, α-linoleic acid, γ-linoleic acid, stearic acid, and arachidic acid. With the change in the development stage, there were significant differences in the content of each lipid; however, the content was mainly concentrated in the middle stage.

Transcriptome sequencing is the basis of gene function and structure research and can reflect the overall characteristics of gene expression regulation in biological processes [59]. In this work, RNA-seq sequencing was performed on 15 samples. The Q20 and Q30 values were greater than 90%, indicating the quality of the sequencing results. All samples produced a combined total of 778,549,924 clean reads and 31,902 DEGs. The PCA showed that the samples in the five periods were obviously divided, indicating that the genes with differential expression levels played a significant regulatory role in the development of kernels with the gradual development and maturation of *P. koraiensis* seeds. In addition, all DEGs were annotated in the KEGG enrichment pathway. The results show that 63 DEGs are enriched in the biosynthesis of the unsaturated fatty acid pathway, and 49 DEGs are enriched in the fatty acid elongation pathway, indicating that these genes are involved in the lipid synthesis process. WGCNA and gene interaction network analysis identified a total of 10 hub containing genes that code for enzymes involved in the formation of α-linolenic acid, linoleic acid, and kernel development. The gene expression heatmap showed that these genes were highly expressed in the later stage while the kernel was undergoing morphogenesis. Therefore, the products of these genes may play a crucial role in the formation of *P. koraiensis* seeds and could influence their characteristics, such as size and weight. The abovementioned results are sufficient to prove that there is a rich lipid biosynthesis process in the late stage of kernel development. Reports on *Jatropha curcas* [60] and *Hippophae* L. [61] also proved that the synthesis of lipids was accelerated in the later stage of seed development. In *J. curcas*, a total of 68 genes involved in fatty acid and lipid biosynthesis were identified by the relative expression level of genes during endosperm development, of which 8 gene expression levels were continuously increased with the development of the endosperm. While the seeds of *Hippophae* L. contain high levels of fatty acids, of which linoleic and α-linolenic acid contents can be as high as 39% or more, the study also showed that some miRNAs may have significant importance in regulating lipid biosynthesis and seed size.

Pyruvate dehydrogenase (PDH) is a crucial rate-limiting enzyme in the de novo synthesis of fatty acids and is the basis of lipid synthesis. Its main function is to catalyze pyruvate to acetyl-CoA [62]. The overexpression of *PDH* genes has been reported to effectively promote the biosynthesis of acetyl-CoA [63]. In this work, a total of 12 *PDH* gene expression levels were determined with high expression levels at the S1 and S5 stages. It can be speculated that they have a crucial regulatory function in the formation of acetyl-CoA at the early and late stages of kernel development. In addition, reports of *Juglans regia* L. and *Carya illinoinensis* have shown that most genes related to fatty acid synthesis have high expression levels during the whole development of the embryo, among which PDH has a better performance [62,64]. The *ACC*, *ACP*, *KAS*, *SAD*, *FATA*, and *FATB* genes identified in this study were expressed at different levels in several stages, indicating the complexity of fatty acid biosynthesis during the development of *P. koraiensis* kernels. Furthermore, TAG is mainly stored in the cytoplasm in the form of oil bodies after synthesis, with oleosin, caleosin, and stero being the major regulators of oil body formation [65]. Oleosin is the main protein that regulates the structure and function of oil bodies [66]. In a study of oil body formation in sesame seeds, three oleosin genes were significantly highly expressed in mature seeds, ensuring the structural stability of the oil body [67,68]. Caleosin is mainly involved in the synthesis and metabolism of oil bodies, and stero is an important carrier of signal transduction in plants [69,70]. It is worth noting that two oleosin genes, five caleosin genes, and one stero gene were identified in this study. These genes were highly expressed in the late stage of development, indicating that oil body accumulation was mainly concentrated there. This finding is consistent with the distribution trend of lipid content in the metabolome data, further proving the accuracy of our research results.

## 4. Materials and Methods

### 4.1. Materials and Sampling

The experimental materials used in this study were collected from the Naozhi *P. koraiensis* seed orchard of the Linjiang Forestry Bureau in Jilin Province (41°80′ N, 126°90′ E) in 2021. In this study, five cone samples of the same clone were collected and divided into five periods (130, 160, 190, 220, and 240 days of the year) according to the phenotypic characteristics (such as size and shape). The seed kernels were isolated from the cones, with three biological replicates per period. For subsequent transcriptome and metabolome sequencing, these samples were immediately frozen in liquid nitrogen and stored in a −80 °C freezer.

### 4.2. Metabolite Extraction and Analysis

A lyophilizer (Scientz-100F) (Xinzhi, Ningbo, China) was used to immediately vacuum freeze-dry the samples, which were then ground for 1.5 min at 30 Hz using a grinder (MM400) (Retsch, Arzberg, Germany). Methanol (1.2 mL) was used to extract 100 mg of powder. The extract was vortexed every 30 min for 30 s for a total of 6 times. The solution was centrifuged at 12,000 rpm for 10 min at 4 °C, and the supernatant was collected and filtered with a microporous membrane (0.22 μm pore size) before UPLC–MS/MS analysis, which was performed by Wuhan MetWare Biotechnology Co., Ltd. (www.mettware.cn, accessed on 10 June 2022) using the methods of Li et al. [71]. To identify the differences between samples, hierarchical analysis (HCA) and orthogonal partial least-squares discriminant analysis (OPLS-DA) were performed on the normalized metabolite data using the R package. Principal component analysis (PCA) using R software (v1.20.0) (www.r-project.org/, accessed on 10 February 2023) [72] built-in statistical prcomp function, set prcomp function parameter scale = True, demonstrated the data unit variance scaling (UV) normalization. The Pearson’s correlation coefficient was calculated using the built-in cor function of R software. The correlation between the two replicate samples was stronger the closer the absolute value of r was to 1.

### 4.3. RNA-Seq and Functional Annotation

RNA isolation, cDNA library construction, and sequencing were executed following the methods described in Zhang’s research [73]. The purity, quantification, and integrity of total RNA were assessed using a NanoPhotometer spectrophotometer (IMPLEN, Westlake Village, CA, USA), Qubit 2.0 Fluorimeter (Life Technologies, Carlsbad, CA, USA), and Agilent 2100 bioanalyzer (Agilent Technologies, Santa Clara, CA, USA). PolyA-mRNA was purified by oligo (dT) magnetic beads. Using a fragmentation buffer, RNA was cleaved into small fragments and, finally, these small fragments were used as templates to synthesize first-strand cDNA. A total of fifteen libraries from the five groups were constructed after the PCR amplification of the cDNA. Then, all libraries were sequenced on the Illumina HiSeq platform after qualification.

To ensure the accuracy of subsequent transcriptome sequencing analysis, fastp software (version 0.12) [74] was used to accurately filter the raw reads and finally obtain clean reads with high quality. Then, the clean reads were spliced using Trinity software [49]. Based on the public database, DIAMOND BLASTX software [75] was used to compare the assembled Unigene sequences with KEGG (Kyoto Encyclopedia of Genes and Genomes), GO (Gene Ontology), Nr (NCBI nonredundant protein sequences), Pram (Protein family), COG (Clusters of Orthologous Group of proteins), Swiss-Prot (a manually annotated and curated protein sequence database), KOG (eukaryotic Ortholog Groups), and Trembl databases to obtain the gene functional annotation information of UniGenes in each database. FPKM was used to evaluate the expression levels of each gene by means of RSEM software [71].

### 4.4. Differentially Expressed Gene (DEG) and GO and KEGG Enrichment Analyses

DESeq2 R package [72] was used to analyze the differential expressions between different groups of samples, and then the Benjamini–Hochberg method was used to perform multiple hypothesis test corrections on the *p*-value to obtain the false discovery rate. The screening conditions for differentially expressed genes were set to |log2Fold Change| ≥ 1 and FDR < 0.05. The GO and KEGG enrichment analysis of the DEGs was implemented by the ChiPlot (https://www.chiplot.online/, accessed on 15 February 2023) online website, and the parameters were set as default values [76].

### 4.5. Weighted Gene Co-Expression Network Analysis (WGCNA)

Weighted gene co-expression network analysis (WGCNA) (https://cloud.metware.cn/#/home, accessed on 6 March 2023) [77] was used to extensively analyze the DEGs and lipid contents of seed kernels at various developmental stages, and gene clusters highly related to lipid biosynthesis were identified. The soft threshold power, min module size, and cut height were 14, 30, and 0.25, respectively.

### 4.6. Gene Interaction Network Analysis

To filter the hub genes in the module, the top hub genes with kWithin values in the modules were examined through a gene interaction network analysis using the String (https://cn.string-db.org/, accessed on 8 March 2023) [78] online website. Then, hub genes were visualized using Cytoscape (v3.9.1) software. Then, the top hub genes were obtained based on the MCC algorithm using the cytoHubba plugin, and a gene expression heatmap was generated with TBtools [79].

### 4.7. Quantitative Real-Time PCR (qRT-PCR) Validation

The specific primers for DEGs in the transcriptome data were designed using the Primer3 online website (https://primer3.ut.ee/, accessed on 15 March 2023) [80], and the primer sequences are shown in Appendix A. Each 20 μL reaction consisted of 0.4 μL of ROX Reference Dye Ⅱ, 10 μL of 2 × SYBR (TB Green) Primix Ex Taq II, 6 μL of double-distilled water (ddH_2_O), 0.8 μL of upstream and downstream primers, and 2 μL of cDNA template. The quantitative real-time PCR (qRT-PCR) validation was tested using the ABI 7500 Real-Time system (Applied Biosystems, Beijing, China), and the reaction procedure consisted of 95 °C for 30 s, 40 cycles of 95 °C for 5 s, 61 °C for 35 s, 95 °C for 15 s, 60 °C for 1 min, and 95 °C for 15 s. The 18S gene (forward primer: GAGGTAGCTTCGGGCGCAACT, and reverse primer: GCAGGTTAGCGAAATGCGATAC) was utilized as a reference gene. Finally, three technical repetitions were executed for the experiment; the relative expression levels were calculated using the 2^−ΔΔCT^ method [71]. The S1 stage was used as a reference sample to calculate the 2-delta-delta Ct value.

## 5. Conclusions

*P. koraiensis* is a precious conifer species with significant economic and ecological worth. Its seed kernels are not only delicious, but also have a high nutritional value. In this study, a combination of widely targeted metabolomics and transcriptomics was used to examine the biosynthesis mechanism of lipids during the development of seed kernels. The results demonstrate that the seed kernels have a high lipid content. A total of 148 lipid metabolites were identified in this study, among which palmitaldehyde, α-linolenic acid, 3-hydroxyoctadecanoic acid, octadeca-11E,13E,15Z-trienoic acid, linoleic acid, elaidic acid, punicic acid (9Z,11Z,13Z-octadecatrienoic acid), myristic acid, 9,10,13-trihydroxy-11-octadecenoic acid, and 11-octadecanoic acid (vaccenic acid) were present in relatively high quantities, accounting for 78.34% of all metabolites. The content of the important lipid metabolites α-linolenic acid and linoleic acid increased with the extension of development time and peaked at the S5 stage, indicating their importance in the later lipid synthesis process. In addition, the transcriptome gene function enrichment analysis showed that a total of 63 DEGs were associated with the biosynthesis of unsaturated fatty acids, and 49 DEGs were involved in the fatty acid elongation process. WGCNA further identified hub genes associated with metabolite and seed kernel development. The results reveal that the 10 hub genes identified using the MCC algorithm are engaged in the biosynthesis of lipids and seed kernel development, and that they present a significant uptrend in the late stages of development, indicating that these genes play a vital role in these processes. The biosynthesis process and critical functional genes of lipids in *P. koraiensis* seed kernels were revealed in this study, combined with extensive targeted metabolomics and transcriptomics analyses. This information served as a reference for further molecular mechanism research as well as the food development and utilization of *P. koraiensis*.

## Figures and Tables

**Figure 1 ijms-24-12887-f001:**
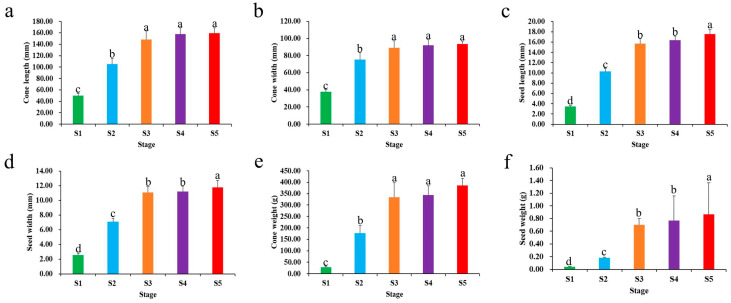
Phenotypic characterization analysis of cone and seed development. (**a**) Length of cone, (**b**) width of cone, (**c**) length of seed, (**d**) width of seed, (**e**) weight of cone, (**f**) weight of seed. Samples at 130, 160, 190, 220, and 240 days of the year as test materials and labeled as S1–S5. The differences analysis was conducted using the IBM SPSS Statistics v26.0 software with the Student–Newman–Keuls multiple range test; error bars represent the SD of the means at n = 30; bars with different lowercase letters are significantly different (*p* < 0.05).

**Figure 2 ijms-24-12887-f002:**
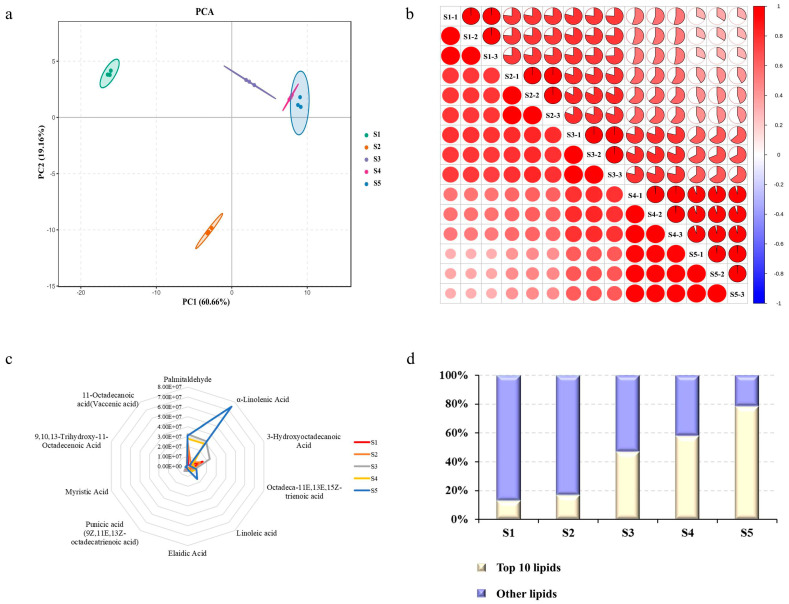
Lipid metabolism analysis of seed kernel development. (**a**) PCA of lipid metabolism in different samples. (**b**) Intragroup correlation of seed samples in different developmental stages. (**c**) Radar chart of the 10 most abundant free fatty acids. (**d**) Ratio of the 10 most abundant free fatty acids to total lipids. Samples at 130, 160, 190, 220, and 240 days of the year as test materials and labeled as S1–S5.

**Figure 3 ijms-24-12887-f003:**
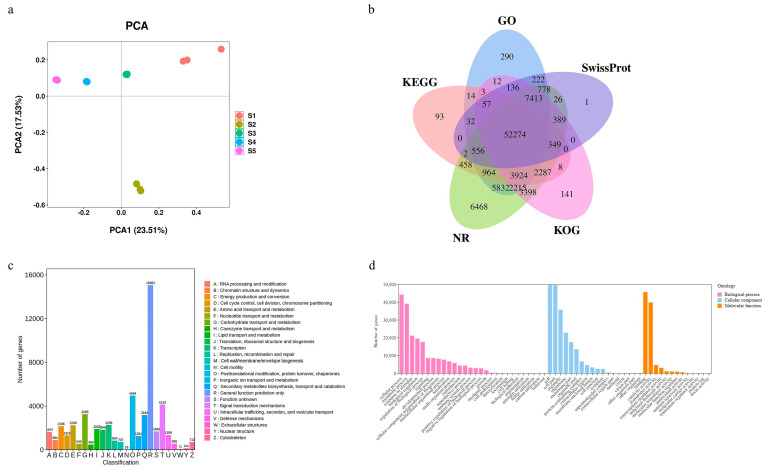
Transcriptome sequencing and gene functional annotation during kernel developmental stages. (**a**) PCA of transcriptome data. (**b**) Venn diagram of annotations from the five major databases. (**c**) KOG functional classification of differentially expressed genes. (**d**) Gene Ontology functional annotation of differentially expressed genes.

**Figure 4 ijms-24-12887-f004:**
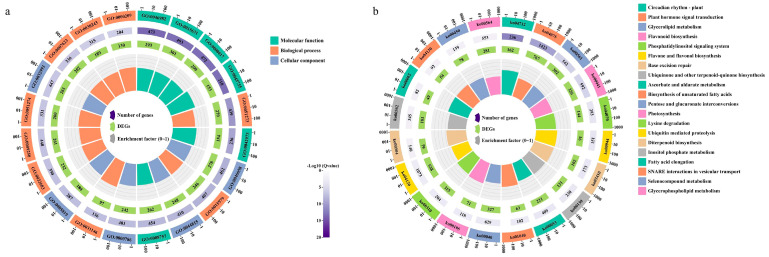
Enrichment circle diagram of differentially expressed genes during *P. koraiensis* seed kernel development in the GO (**a**) and KEGG (**b**) databases. There are four circles from outside to inside: the first circle represents the classification of enrichment and the outside circle serves as the coordinate ruler for the number of DEGs. The second circle shows the number of classifications and Q (*p*) values in the background DEGs. The bar lengthens with increasing DEGs and turns purple with a decreasing Q value. The bar graph of annotated DEGs is shown in the third circle. The fourth circle exhibits the enrichment factor value of each category, and each cell of the background auxiliary line represents 0.1.

**Figure 5 ijms-24-12887-f005:**
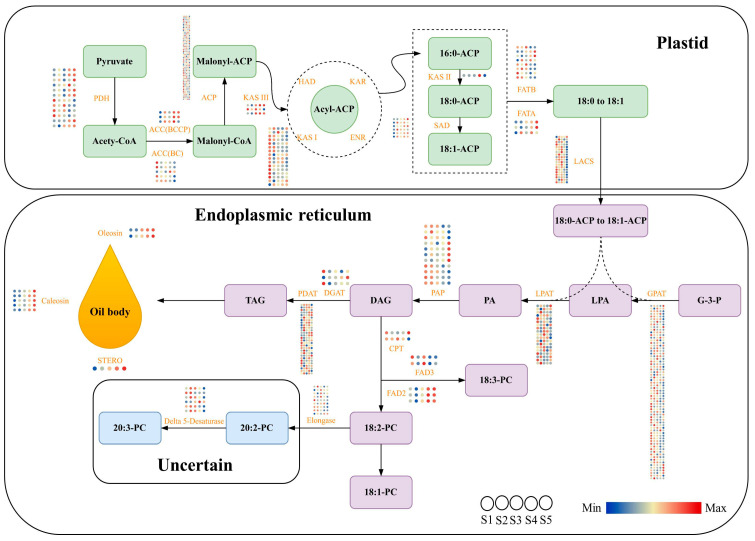
Identification of differentially expressed genes *P. koraiensis* seed kernel development involved in the lipid biosynthesis synthesis pathway. PDH, pyruvate dehydrogenase; ACC (BCCP), biotin carboxyl carrier protein of heteromeric acetyl-CoA carboxylase; ACC (BC), biotin carboxylase subunit of heteromeric acetyl-CoA carboxylase; ACP, acyl-carrier protein; KAS, ketoacyl-ACP synthase; HAD, 3-hydroxyacyl-CoA dehydrogenase; KAR, β-ketoacyl-ACP reductase; ENR, enoyl-ACP reductase; SAD, stearoyl-ACP reductase; FATA, acyl-ACP thioesterase A; FATB, acyl-ACP thioesterase B; LACS, long-chain acyl-CoA synthetase; GPAT, glycerol-3-phosphate acyltransferase; LPAT, lysophosphatidic acid acyltransferase; PAP, phosphatidic acid phosphatase; DGAT, diacylglycerol acyltransferase; PDAT, phospholipid: diacylglycerol acyltransferase; STERO, steroleosin; CPT, diacylglycerol cholinephosphotransferase; FAD2, fatty acid desaturase-2; FAD3, fatty acid desaturase-3. The min (green) to max (red) color scale refers to the range of gene expression levels from low to high. Samples at 130, 160, 190, 220, and 240 days of the year as test materials and labeled as S1–S5.

**Figure 6 ijms-24-12887-f006:**
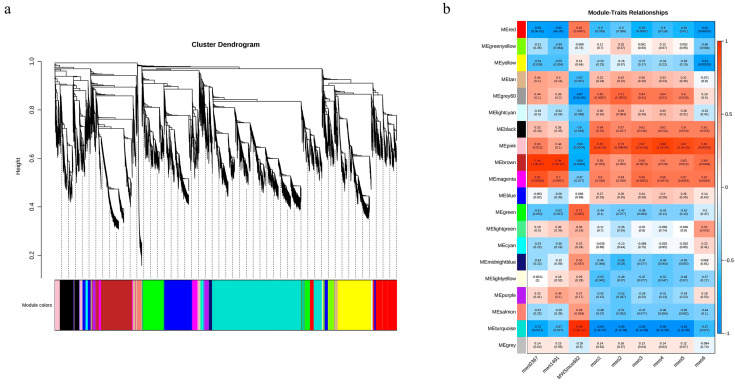
Weighted gene co-expression network analysis during seed kernel development. (**a**) Clustering dendrograms of genes and module division. (**b**) Correlation heatmap between modules and traits. The indicators are as follows: mws1 (cone length), mws2 (cone width), mws3 (seed length), mws4 (seed width), mws5 (cone weight), mws6 (seed weight), mws0367 (α-linolenic acid), mws1491 (linoleic acid), and MWSmce692 (γ-linolenic acid).

**Figure 7 ijms-24-12887-f007:**
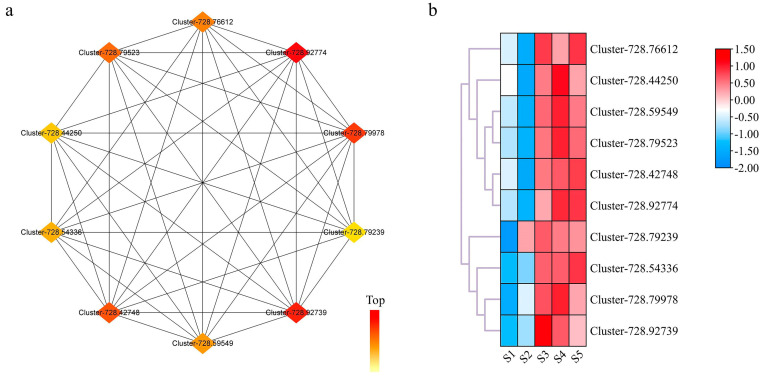
Regulatory network of lipid synthesis genes based on the pink module. (**a**) The top 10 hub genes in the pink module. (**b**) Expression analysis of the top 10 hub genes. The min (blue) to max (red) color scale refers to the range of gene expression levels from low to high, which represents log2-transformed FPKM. Gene expression was normalized using Z-scores of fragments per kilobase of exon per million fragments mapped (FPKM) as an average. Samples at 130, 160, 190, 220, and 240 days of the year as test materials and labeled as S1–S5.

## Data Availability

All transcriptome raw data were submitted to the NCBI database under accession number PRJNA910958, and metabolome data were submitted to the MetaboLights under accession number MTBLS8257.

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
