# Peer review of "Widely Targeted Metabolomic Profiling Combined with Transcriptome Analysis Provides New Insights into Lipid Biosynthesis in Seed Kernels of Pinus koraiensis"

_ijms, 2023, doi:10.3390/ijms241612887_

Round 1

Reviewer 1 Report

Manuscript entitled “Widely targeted metabolomic profiling combined with transcriptome analysis provides new insights into lipid biosynthesis in seed kernels of Pinus koraiensis” by Li et al describes a study of characterization of the changes in metabolism and gene expression during the development of seed kernels of Pinus koraiensis. The authors performed the determination of transcriptomes (RNAseq) and lipid contents (identified with UPLC‒MS/MS) of samples of cone seeds harvested during 5 developmental stages (named S1-S5).  With the data obtained the authors performed in silico comparative analyses using several data analysis software and gene expression was also checked by qRT-PCR. The experimental design and the proposed approaches used in this work seem both correct for the most part. Also, the results obtained are interesting. However, in my opinion, some aspects of the manuscript need to be revised, particularly how the results are presented, described, and discussed, before considering this work suitable for publication. Some of them are enumerated in the additional comments below.

Main issues:

a)      Supplementary material is absent, so it is difficult to assess whether the data cited in the text is adequate or presented correctly.

b)     Figures resolution should be improved. Most of them are barely readable.

c)      Regarding DEGs, the authors should describe better which kind of sample comparisons have been performed.

d)     The discussion of the results obtained is often rather superficial and should be improved. Although the authors have performed a good interpretation of the results obtained, the discussion consists in some parts of a mere repetition of the description of the data obtained (that has already been detailed in the results section or SHOULD be there) and often lacks integration into previous results from the literature that could help fit their observations in a more robust framework.

Additional comments:

Regarding Introduction

Line 79:  The term “the nutrition” is not appropriate to refer to the plant's hydrocarbon molecules stored as reserves.

Line 82: “Genes” do not promote anything…they encode information for the synthesis of macromolecules (tRNAs, rRNAs, proteins…) that perform functions in living organisms.

Line 83-84: Polyamines are not considered amino acids. Also, polyamine abbreviation (PA) should be defined in this sentence, but only appears one time in the main text, so I doubt its utility. The same could be applied to other abbreviations used in the manuscript, like ABA in line 85. Please check all the abbreviations and only use them when necessary.

Line 97: Which is the starting point of cone age? Pollination? This should be indicated here and, in the material and methods section.

Line 98: The criteria used to define the cone developmental stages s1-S5 should be described or referenced, at least in the material and methods section.

Line 102: Data shown in Figure 1 should include a statistical analysis that could help to a future reader identify the significant changes in each parameter.

Line 108-109: Figure S1 is absent.

Line 110: Table S2 is absent. Also, there is no previous mention of Table S1 in the main text, so…the table numbering is wrong.

Lines 119-120: Which kind of sequencing data? All the data present in Figure 2 is metabolomic.

Line 125: please correct “the their lipids content” I am not sure if you are refereeing to the 10 most abundant free fatty acids or not…

Line 139: table S3 is absent. Also, “Trinity software (reference)”

Lines 143-144: comprising 74722 annotated in the GO database” and so on: “61021 annotated”, etc.

Lines 152-155: As a reader, I do not see the relevance of describing that “Cell” was the most enriched term in “cellular component” GO category. The same for “cellular process” in the “biological process” category since both are the top categories in their respective GO hierarchies. In other words, this is the expected result for any transcriptome annotation….

Line 156: Figure 3 resolution should be improved. GO functional annotation terms in figure 4d are illegible.

Line 162: DEG numbers should be detailed to a future reader, which stade or sample was used as a reference? Which criteria was used to choose DEGs? Figure 4 refers to which set of DEGs?

Line 169: Figure S2 is absent.

Lines 182-189: Figure 4: Resolution should be improved, because number ansd letters are barely readable. Also,  “rich factor” should be changed to “enrichment factor” in both figures and figure legend. On the other hand, I think it would be clearer if the circular diagram of figure 4a were decomposed into three diagrams, (Molecular function, Biological process, and Cellular component). In this way, it would be easier for a future reader to identify the most enriched functional categories for those three main GO categories.

Lines 197-198: ACC (BC), 197 KAS â… , ACP and LACS abbreviatures should be defined.

Line 200: ACC (BCCP), SAD and PHD abbreviatures should be defined.

Line 202: GPAT, LPAT and 202 PDAT abbreviations should be defined.

Line 204: STERO abbreviation should be defined.

Line 232: Figure 6 resolution should be increased: numbers and letters in figure 6b are barely readable.

Line 246-249: The heatmap represented in figure 7b represents relative gene expression… which sample/stage was used as reference? This information should be described in the figure legend.

Line 254: Figure S3 is absent.

Regarding Discussion

Line 278 : Metabolomics and sequencing are two different techniques…please correct

Lines 187-288: Similarities and differences of the results obtained by Li et at (2021) should be detailed.

Lines 297-300: Please be more specific in the description of the results obtained: “a set of 63 DEGs was enriched in genes coding for enzymes of the biosynthesis of the unsaturated fatty acid pathway”. The same for the ser of 49 DEGs. Please correct.

Line 302: “a total of 10 hubs containing genes that code for enzymes involved…”

Line 304: the gene expression heatmap by itself could not demonstrate that “these genes determined the size, character, and weight of the seeds.” I kindly urge the authors to be more cautious with their affirmations.

Lines 306-307: Again, the similarities with the studies on Jatropha curcas and Sea buckthorn should be described and discussed in more depth. Also, Sea buckthorn is not the scientific name of the plant species.

Lines 310, 311: PDH should appear in italics (you are talking of the gene)

Lines 311-312:  The purpose of transcriptome studies is not the “obtention” of any gene, but the determination of gene expression levels. So, the sentence “In this work, a total of 12 PDH genes were obtained with high expression levels at the S1 and S5 stages” should be rewritten appropriately.

Line 313: Again, genes are DNA segments that code for proteins. Therefore, in the sentence "They catalyze the formation of acetyl-CoA at the early and late stages of kernel development" it should be better to indicate who it refers to.

Line 314: results in references 61 and 63 should be discussed in more detail.

Line 315:” The ACC, 315 ACP, KAS, SAD, FATA and FATB genes”, gene names should appear in italics.

Line 367: “Trinity software”.

Line 377: DESeq2 software should be referenced.

Lines 377-382: The stage or sample used as a reference and which samples were compared against others to determine DEGs should be described better.

Regarding Material and Methods

Line 335: days starting from which point in particular?

Line 381: ChiPlot software should be referenced, if available.

Line 385: Software used to perform WGCNA analyses should be detailed or referenced.

Line 391: String software should be referenced, if available

Line 397: Primer3 software should be referenced.

Line 398: Table S1 is absent.

Line 401-406: The methodology for obtaining the relative expression levels from the Ct data of qRT-PCR must be described or referenced.

Regarding data availability

Lines 441-441: Metabolome data should be also uploaded to an appropriate public database.

Regarding Supplementary material

Supplementary material is absent.

Regarding References:

Please double check your list of references, some contain the full name of the journal, others abbreviate name. Please unify them according to the IJMS required format.

The manuscript demonstrates mostly good English writing and grammar

Author Response

List of responses

Manuscript ID: ijms-2480670

Widely targeted metabolomic profiling combined with transcriptome analysis provides new insights into lipid biosynthe-sis in seed kernels of Pinus koraiensis

Dear Editor-in-chief and Reviewers:

Thank you for your letter and for the reviewers’ comments concerning our manuscript entitled “Widely targeted metabolomic profiling combined with transcriptome analysis provides new insights into lipid biosynthe-sis in seed kernels of Pinus koraiensis” (ijms-2480670). Those comments are all valuable and very helpful for revising and improving the quality our paper. We have studied all comments carefully and have made corrections which we hope meet the standard of your highly esteemed journal. For ease of tracking, we highlighted the changes with red color in the revised version. The main corrections in the paper and the responses to the reviewers’ comments are given below.

Responds to reviewer’s comments:

Reviewer #1

Manuscript entitled “Widely targeted metabolomic profiling combined with transcriptome analysis provides new insights into lipid biosynthesis in seed kernels of Pinus koraiensis” by Li et al describes a study of characterization of the changes in metabolism and gene expression during the development of seed kernels of Pinus koraiensis. The authors performed the determination of transcriptomes (RNAseq) and lipid contents (identified with UPLC‒MS/MS) of samples of cone seeds harvested during 5 developmental stages (named S1-S5). With the data obtained the authors performed in silico comparative analyses using several data analysis software and gene expression was also checked by qRT-PCR. The experimental design and the proposed approaches used in this work seem both correct for the most part. Also, the results obtained are interesting. However, in my opinion, some aspects of the manuscript need to be revised, particularly how the results are presented, described, and discussed, before considering this work suitable for publication. Some of them are enumerated in the additional comments below.

Main issues:

  1. a) Supplementary material is absent, so it is difficult to assess whether the data cited in the text is adequate or presented correctly.

  1. b) Figures resolution should be improved. Most of them are barely readable.

  1. c) Regarding DEGs, the authors should describe better which kind of sample comparisons have been performed.

  1. d) The discussion of the results obtained is often rather superficial and should be improved. Although the authors have performed a good interpretation of the results obtained, the discussion consists in some parts of a mere repetition of the description of the data obtained (that has already been detailed in the results section or SHOULD be there) and often lacks integration into previous results from the literature that could help fit their observations in a more robust framework.

Major

1) Line 79: The term “the nutrition” is not appropriate to refer to the plant's hydrocarbon molecules stored as reserves.

Response: The word “the nutrition” has been replaced by “useful component” in the sentence (see line 79).

2) Line 82: “Genes” do not promote anything…they encode information for the synthesis of macromolecules (tRNAs, rRNAs, proteins…) that perform functions in living organisms.

Response: The sentences have been rephrased (see line 82).

3) Line 83-84: Polyamines are not considered amino acids. Also, polyamine abbreviation (PA) should be defined in this sentence, but only appears one time in the main text, so I doubt its utility. The same could be applied to other abbreviations used in the manuscript, like ABA in line 85. Please check all the abbreviations and only use them when necessary.

Response: The sentences have been rephrased (see line 84-85).

4) Line 97: Which is the starting point of cone age? Pollination? This should be indicated here and, in the material and methods section.

Response: The starting point of cone age is calculated from the first day of this year. It was indicated in line 102.

5) Line 98: The criteria used to define the cone developmental stages s1-S5 should be described or referenced, at least in the material and methods section.

Response: It was described in line 384-386.

6) Line 102: Data shown in Figure 1 should include a statistical analysis that could help to a future reader identify the significant changes in each parameter.

Response: Thanks. This study has supplemented the statistical analysis, see Figure 1.

7) Line 108-109: Figure S1 is absent.

Response: Very sorry. Due to my negligence, the Supplementary Figures has been uploaded to the journal.

8) Line 110: Table S2 is absent. Also, there is no previous mention of Table S1 in the main text, so…the table numbering is wrong.

Response: Very sorry. Due to my negligence, the Supplementary Tables has been uploaded to the journal.

9) Lines 119-120: Which kind of sequencing data? All the data present in Figure 2 is metabolomic.

Response: The sequencing data refers to metabolomic, which has been rephrased (see line 124).

10) Line 125: please correct “the their lipids content” I am not sure if you are refereeing to the 10 most abundant free fatty acids or not…

Response: The sentences have been corrected (see line 129).

11) Line 139: table S3 is absent. Also, “Trinity software (reference)”

Response: Very sorry, due to my negligence, the Supplementary Tables has been uploaded to the journal. Trinity software was referenced in line 417.

12) Lines 143-144: comprising 74722 annotated in the GO database” and so on: “61021 annotated”, etc.

Response: The sentences have been rephrased (see line 153-154).

13) Lines 152-155: As a reader, I do not see the relevance of describing that “Cell” was the most enriched term in “cellular component” GO category. The same for “cellular process” in the “biological process” category since both are the top categories in their respective GO hierarchies. In other words, this is the expected result for any transcriptome annotation….

Response: The sentence have been deleted (see line 161).

14) Line 156: Figure 3 resolution should be improved. GO functional annotation terms in figure 4d are illegible.

Response: Figure 3 and 4 resolution have been improved in this manuscript, and we reuploaded all high-resolution figures.

15) Line 162: DEG numbers should be detailed to a future reader, which stade or sample was used as a reference? Which criteria was used to choose DEGs? Figure 4 refers to which set of DEGs?

Response: The criteria for choosing DEGs have been indicated in the line 437-439. And the DEGs in Figure 4 refers to common DEGs in all groups.

16) Line 169: Figure S2 is absent.

Response: Very sorry. Due to my negligence, the Supplementary Figures has been uploaded to the journal.

17) Lines 182-189: Figure 4: Resolution should be improved, because number ansd letters are barely readable. Also, “rich factor” should be changed to “enrichment factor” in both figures and figure legend. On the other hand, I think it would be clearer if the circular diagram of figure 4a were decomposed into three diagrams, (Molecular function, Biological process, and Cellular component). In this way, it would be easier for a future reader to identify the most enriched functional categories for those three main GO categories.

Response: Figure 4 resolution has been improved in this manuscript, and the word “rich factor” has been replaced by “enrichment factor” in the sentence (see line 209 and Figure 4). Figure 4a were decomposed into three diagrams, such as Molecular function, Biological process, and Cellular component in Figure S2.

18) Lines 197-198: ACC (BC), 197 KAS â… , ACP and LACS abbreviatures should be defined.

Response: All abbreviatures have been defined (see line 218-221).

19) Line 200: ACC (BCCP), SAD and PHD abbreviatures should be defined.

Response: All abbreviatures have been defined (see line 223-224).

20) Line 202: GPAT, LPAT and 202 PDAT abbreviations should be defined.

Response: All abbreviatures have been defined (see line 227-228).

21) Line 204: STERO abbreviation should be defined.

Response: All abbreviatures have been defined (see line 230).

22) Line 232: Figure 6 resolution should be increased: numbers and letters in figure 6b are barely readable.

Response: Figure 6 resolution has been improved in this manuscript.

23) Line 246-249: The heatmap represented in figure 7b represents relative gene expression… which sample/stage was used as reference? This information should be described in the figure legend.

Response: Although supplementary explanation have been described in the figure legend, but I did not understand the meaning of the sentence “which sample/stage was used as reference?” (see line 275-277). This study is only a heat map drawn from the expression levels of the 10 genes screened at 5 developmental stages, and no reference is required.

24) Line 254: Figure S3 is absent.

Response: Very sorry. Due to my negligence, the Supplementary Figures has been uploaded to the journal.

25) Line 278 : Metabolomics and sequencing are two different techniques…please correct

Response: The sentences have been corrected (see line 305).

26) Lines 187-288: Similarities and differences of the results obtained by Li et at (2021) should be detailed.

Response: This content has been described in detail in the text (see line 318-321).

27) Lines 297-300: Please be more specific in the description of the results obtained: “a set of 63 DEGs was enriched in genes coding for enzymes of the biosynthesis of the unsaturated fatty acid pathway”. The same for the ser of 49 DEGs. Please correct.

Response: The content (The results showed that 63 DEGs were enriched in the biosynthesis of the unsaturated fatty acid pathway, and 49 DEGs were enriched in the fatty acid elongation pathway, indicating that these genes were involved in the lipid synthesis process) has been corrected (see line 331-332).

28) Line 302: “a total of 10 hubs containing genes that code for enzymes involved…”

Response: The sentences have been corrected (see line 334).

29) Line 304: the gene expression heatmap by itself could not demonstrate that “these genes determined the size, character, and weight of the seeds.” I kindly urge the authors to be more cautious with their affirmations.

Response: The content have been corrected (see line 337-338).

30) Lines 306-307: Again, the similarities with the studies on Jatropha curcas and Sea buckthorn should be described and discussed in more depth. Also, Sea buckthorn is not the scientific name of the plant species.

Response: This content has been described in detail in the text, and the word “Sea buckthorn” has been replaced by “Hippophae L.” in the sentence (see line 340-347).

31) Lines 310, 311: PDH should appear in italics (you are talking of the gene)

Response: The PDH have been changed into an italics (see line 350).

32) Lines 311-312: The purpose of transcriptome studies is not the “obtention” of any gene, but the determination of gene expression levels. So, the sentence “In this work, a total of 12 PDH genes were obtained with high expression levels at the S1 and S5 stages” should be rewritten appropriately.

Response: The content have been corrected (see line 352).

33) Line 313: Again, genes are DNA segments that code for proteins. Therefore, in the sentence "They catalyze the formation of acetyl-CoA at the early and late stages of kernel development" it should be better to indicate who it refers to.

Response: The content have been corrected (see line 353).

34) Line 314: results in references 61 and 63 should be discussed in more detail.

Response: This content has been described in detail in the text (see line 354-357).

35) Line 315:” The ACC, 315 ACP, KAS, SAD, FATA and FATB genes”, gene names should appear in italics.

Response: These genes have been changed into an italics (see line 357).

36) Line 367: “Trinity software”.

Response: The word “Trinity software” has been added in the sentence (see line 417).

37) Line 377: DESeq2 software should be referenced.

Response: DESeq2 software have been referenced (see line 435).

38) Lines 377-382: The stage or sample used as a reference and which samples were compared against others to determine DEGs should be described better.

Response: This content has been described in the text.

39) Line 335: days starting from which point in particular?

Response: The days starting from the first day in this year, this content has been added in the text (see line 385).

40) Line 381: ChiPlot software should be referenced, if available.

Response: ChiPlot software has been referenced in the text (see line 440).

41) Line 385: Software used to perform WGCNA analyses should be detailed or referenced.

Response: Software used to perform WGCNA analyses has been referenced in the text (see line 444).

42) Line 391: String software should be referenced, if available

Response: String software has been referenced in the text (see line 451).

43) Line 397: Primer3 software should be referenced.

Response: String software has been referenced in the text (see line 457).

44) Line 398: Table S1 is absent.

Response: Very sorry. Due to my negligence, the Supplementary Tables has been uploaded to the journal.

45) Line 401-406: The methodology for obtaining the relative expression levels from the Ct data of qRT-PCR must be described or referenced.

Response: The methodology for obtaining the relative expression levels from the Ct data of qRT-PCR has been described (see line 465-467).

46) Lines 441-441: Metabolome data should be also uploaded to an appropriate public database.

Response: Very sorry, the original data of the metabolome is being uploaded to the MetaboLights database of the European Institute of Bioinformatics (https://www.ebi.ac.uk/metabolights/) with the accession number MTBLS8257, and it will take some time to complete.

47) Supplementary material is absent.

Response: All supplementary materials have been re-uploaded to the magazine.

48) Please double check your list of references, some contain the full name of the journal, others abbreviate name. Please unify them according to the IJMS required format.

Response: All references have been unified according to the IJMS required format.

Reviewer 2 Report

In the paper entitled "Widely targeted metabolic profiling combined with transcriptome analysis provides new insights into lipid biosynthesis in seed kernels of Pinus koraiensis", the authors study the lipid metabolism of kernels of Pinus koraiensis by RNA-Seq, UPLC‒MS/MS and so on. The design and results of the experiments are beyond doubt. The methods are described quite clearly and reproducibly. This manuscript undoubtedly deserves publication, but I still have some questions.

In the main issue is the lack of supplemental material. I am sure that this is a banal inattention made when sending the manuscript.

Line 97. Why did you use these time points in 130, 160, 190, 220 and 240 days? Was it justified by morphological development or were these random dots?

Figure 1. How much cone and seed was measured to obtain the presented data? It would also be good to evaluate the reliability of the results using the t-test Student.

Line 125. In this case, it would be correct to use "increase", not "rose" Figure 3. The image quality needs to be improved, since the inscriptions are poorly readable.

Line 194. P. koraiensis. Italics.

Author Response

List of responses

Manuscript ID: ijms-2480670

Widely targeted metabolomic profiling combined with transcriptome analysis provides new insights into lipid biosynthe-sis in seed kernels of Pinus koraiensis

Dear Editor-in-chief and Reviewers:

Thank you for your letter and for the reviewers’ comments concerning our manuscript entitled “Widely targeted metabolomic profiling combined with transcriptome analysis provides new insights into lipid biosynthe-sis in seed kernels of Pinus koraiensis” (ijms-2480670). Those comments are all valuable and very helpful for revising and improving the quality our paper. We have studied all comments carefully and have made corrections which we hope meet the standard of your highly esteemed journal. For ease of tracking, we highlighted the changes with red color in the revised version. The main corrections in the paper and the responses to the reviewers’ comments are given below.

Responds to reviewer’s comments:

Reviewer #2

In the paper entitled "Widely targeted metabolic profiling combined with transcriptome analysis provides new insights into lipid biosynthesis in seed kernels of Pinus koraiensis", the authors study the lipid metabolism of kernels of Pinus koraiensis by RNA-Seq, UPLC‒MS/MS and so on. The design and results of the experiments are beyond doubt. The methods are described quite clearly and reproducibly. This manuscript undoubtedly deserves publication, but I still have some questions.

In the main issue is the lack of supplemental material. I am sure that this is a banal inattention made when sending the manuscript.

1) Line 97. Why did you use these time points in 130, 160, 190, 220 and 240 days? Was it justified by morphological development or were these random dots?

Response: In this study, five cone samples of the same clone were collected and divided into five pe-riods (130, 160, 190, 220, and 240 days in this year) according to the phenotypic charac-teristics (such as size and shape).

2) Figure 1. How much cone and seed was measured to obtain the presented data? It would also be good to evaluate the reliability of the results using the t-test Student.

Response: In this study, 30 cones and 30 seeds were measured, and the t-test Student has been supplemented in the content.

3) Line 125. In this case, it would be correct to use "increase", not "rose" Figure 3. The image quality needs to be improved, since the inscriptions are poorly readable.

Response: The word “rose” has been replaced by “increase” in the sentence (see line 130).

4) Line 194. P. koraiensis. Italics.

Response: The word “P. koraiensis” has been italicized (see line 215).

Round 2

Reviewer 1 Report

In this second version of the manuscript the authors have addressed some of the issues detected. However, I must point out that many of them remain unsatisfactorily corrected and therefore I must again recommend a major revision of the document.

Regarding Introduction

Lines 71-78 Again, I think that “useful component” is not a good term. Perhaps something simpler like “The primary source of energy for plants to transition from vegetative growth to are the reserves stored in branches and needles.”

Lines 82-83: The sentence rephrased by the authors remain still incorrect: Genes are not structures with roles in cells or living organisms…. their products perform those tasks: “For instance, the tomato (Solanum lycopersicum) enzymes choline (COD) and betaine aldehyde dehydrogenase (BADH) play a key role in the production of glycinebetaine, which expedites the development of tomato fruits”

Lines 83-84: “Also, polyamines can control strawberry (Fragaria ananassa) fruit development by balancing the levels of ethylene, auxin, and abscisic acid (ABA)”

Regarding Results

Lines 97-98: “"We collected samples at days of the year 130, 160, 190, 220, and 240 as test materials and named them as S1-S5, according to variations in development time."

Lines 104-105, Figure 1 legend: The S1-S5 stages should be defined in the figure legend. Additionally, the meaning of the letters corresponding to the statistical analysis should be indicated. Remember that a figure legend should contain all the information necessary for a future reader to interpret the figure without resorting to the main text.

Line 111: Please Change the table numbering in the supplemental material.  Table S1 should not be the table of primers.

Line 127 “increased significantly.”

Lines 129-132, figure 2 legend:

Line 140:  Again, Please Change the table numbering in the supplemental material.  Table S2 should not be the metabolomic lipid data. Also please replace “Trinity” with ”Trinity software” and reference it here and in line 389 by using the appropriate numbering.

Lines 144-146: I think that ending with an “etc” it is not the best way to describe your results.

“A total of 88342 unigenes were annotated, including 74,722 with GO annotations, 61021 with KEGG terms, 8733 with NR database terms, as well as 47468 with KOG annotations, and 62235 with SwissProt database terms”

Lines 185-185, figure 4 legend: “Enrichment circle diagram of differentially expressed genes during Pinus koraiensis seed kernel development in the GO (a) and 185 KEGG (b) databases”

Lines 215-216, figure 5 legend: “Identification of differentially expressed genes Pinus koraiensis seed kernel development involved in the lipid biosynthesis synthesis pathway.” Also, S1-S5 stages should be defined in figure 5 legend.

Lines 253-257: S1-S5 stages should be defined in figure 7 legend.

Lines 313-316: “The gene expression heatmap showed that these genes were highly expressed in the later stage while the kernel was undergoing morphogenesis. Therefore, the products of these genes may play a crucial role in the formation of P. koraiensis seeds and could influence their characteristics, like size and weight”

Regarding Material and Methods:

Line 356: “days of the year 130, 160, 190, 220, and 240”

Lines 359-360: “"seed kernels were isolated from the cones, with three biological replicates per period."

Lines 399-405:

Like I have mentioned previously for trinity software. References of DESeq2 R package and ChiPlot should be numbered accordingly to their appearance in the main text. The rest of references that appear after them should be also numbered accordingly. I urge the authors to exhaustively review this type of detail regarding the bibliographical references in the entire manuscript.

Line 420: Primer3 software remains unreferenced. Please check the appropriate reference on the website. I think that the original authors of this software will be happy with a recognition. The same could be applied to other web sources used in this work.

Line 422: Please Change the table numbering in the supplemental material.  Table S3 is currently the raw transcriptome statistics data.

Line 431: 2−ΔΔCT method should be referenced. Also, the sample used as a reference to calculate the relative expression data should be indicated.

Lines 467-468: Metabolome data accession in MetaboLights database is still absent.

Regarding figures:

The resolution of figures 2, 3, 4, 5, and 6 is still of poor quality in the PDF version of the manuscript. Please be advised.

Regarding Supplemental Material

Figure S2 Legend: “Gene Ontology enrichment analysis of differentially expressed genes during development of Pinus koraiensis seed kernels.”

Figure S3: RT-qPCR data needs statistical analysis. Significant changes in gene expression should be indicated in the graphs.  

Figure S3 legend: S1-S5 stages should be defined in figure legend. The sample used as a reference to calculate the relative expression data should also be indicated.

I have made some suggestions to the authors, check them.

Author Response

List of responses

Manuscript ID: ijms-2480670

Widely targeted metabolomic profiling combined with transcriptome analysis provides new insights into lipid biosynthe-sis in seed kernels of Pinus koraiensis

Dear Editor-in-chief and Reviewers:

Thank you for your letter and for the reviewers’ comments concerning our manuscript entitled “Widely targeted metabolomic profiling combined with transcriptome analysis provides new insights into lipid biosynthe-sis in seed kernels of Pinus koraiensis” (ijms-2480670). Those comments are all valuable and very helpful for revising and improving the quality our paper. We have studied all comments carefully and have made corrections which we hope meet the standard of your highly esteemed journal. For ease of tracking, we highlighted the changes with red color in the revised version. The main corrections in the paper and the responses to the reviewers’ comments are given below.

Responds to reviewer’s comments:

Reviewer #1

In this second version of the manuscript the authors have addressed some of the issues detected. However, I must point out that many of them remain unsatisfactorily corrected and therefore I must again recommend a major revision of the document.

1) Lines 71-78 Again, I think that “useful component” is not a good term. Perhaps something simpler like “The primary source of energy for plants to transition from vegetative growth to are the reserves stored in branches and needles.”

Response: According to the opinion and reminder of the reviewer, the sentences have been rephrased (see line 77-79).

2) Lines 82-83: The sentence rephrased by the authors remain still incorrect: Genes are not structures with roles in cells or living organisms…. their products perform those tasks: “For instance, the tomato (Solanum lycopersicum) enzymes choline (COD) and betaine aldehyde dehydrogenase (BADH) play a key role in the production of glycinebetaine, which expedites the development of tomato fruits”

Response: According to the opinion and reminder of the reviewer, the sentences have been rephrased (see line 82).

3) Lines 83-84: “Also, polyamines can control strawberry (Fragaria ananassa) fruit development by balancing the levels of ethylene, auxin, and abscisic acid (ABA)”

Response: According to the opinion and reminder of the reviewer, the sentences have been rephrased (see line 83-85).

4) Lines 97-98: “"We collected samples at days of the year 130, 160, 190, 220, and 240 as test materials and named them as S1-S5, according to variations in development time."

Response: According to the opinion and reminder of the reviewer, the sentences have been rephrased (see line 97-98).

5) Lines 104-105, Figure 1 legend: The S1-S5 stages should be defined in the figure legend. Additionally, the meaning of the letters corresponding to the statistical analysis should be indicated. Remember that a figure legend should contain all the information necessary for a future reader to interpret the figure without resorting to the main text.

Response: The S1-S5 stages have been defined in the figure legend. Besides, as much information as possible has been explained in the figure legend (see line 104-108).

6) Line 111: Please Change the table numbering in the supplemental material.  Table S1 should not be the table of primers.

Response: The table numbering in the supplemental material have been changed (see line 114).

7) Line 127 “increased significantly.”

Response: The sentences have been rephrased (see line 129).

8) Lines 129-132, figure 2 legend:

Response: The necessary information has been added to the legend of Figure 2 (see line 134-135).

9) Line 140:  Again, Please Change the table numbering in the supplemental material.  Table S2 should not be the metabolomic lipid data. Also please replace “Trinity” with ”Trinity software” and reference it here and in line 389 by using the appropriate numbering.

Response: The table numbering in the supplemental material have been changed (see line 143). The word “Trinity” has been replaced by “Trinity software” in the sentence (see line 143-144).

10) Lines 144-146: I think that ending with an “etc” it is not the best way to describe your results.

Response: According to the opinion and reminder of the reviewer, the sentences have been rephrased (see line 147-149).

11) Lines 185-185, figure 4 legend: “Enrichment circle diagram of differentially expressed genes during Pinus koraiensis seed kernel development in the GO (a) and 185 KEGG (b) databases”

Response: According to the opinion and reminder of the reviewer, the sentences have been rephrased (see line 186-187).

12) Lines 215-216, figure 5 legend: “Identification of differentially expressed genes Pinus koraiensis seed kernel development involved in the lipid biosynthesis synthesis pathway.” Also, S1-S5 stages should be defined in figure 5 legend.

Response: According to the opinion and reminder of the reviewer, the sentences have been rephrased (see line 217). And S1-S5 stages have been defined in figure 5 legend (see line 228-229).

13) Lines 253-257: S1-S5 stages should be defined in figure 7 legend.

Response: According to the opinion and reminder of the reviewer, the sentences have been rephrased (see line 260-261).

14) Lines 313-316: “The gene expression heatmap showed that these genes were highly expressed in the later stage while the kernel was undergoing morphogenesis. Therefore, the products of these genes may play a crucial role in the formation of P. koraiensis seeds and could influence their characteristics, like size and weight”

Response: According to the opinion and reminder of the reviewer, the sentences have been rephrased (see line 318-320).

15) Line 356: “days of the year 130, 160, 190, 220, and 240”

Response: According to the opinion and reminder of the reviewer, the sentences have been rephrased (see line 357-359).

16) Lines 359-360: “"seed kernels were isolated from the cones, with three biological replicates per period."

Response: According to the opinion and reminder of the reviewer, the sentences have been rephrased (see line 359-360).

17) Like I have mentioned previously for trinity software. References of DESeq2 R package and ChiPlot should be numbered accordingly to their appearance in the main text. The rest of references that appear after them should be also numbered accordingly. I urge the authors to exhaustively review this type of detail regarding the bibliographical references in the entire manuscript.

Response: DESeq2 R package and ChiPlot are not mentioned in the main text, and we have cited them in the material method (see line 374, 400, and 405-406).

18) Line 420: Primer3 software remains unreferenced. Please check the appropriate reference on the website. I think that the original authors of this software will be happy with a recognition. The same could be applied to other web sources used in this work.

Response: According to the opinion and reminder of the reviewer, Primer3 software have been referenced (see line 422).

19) Line 422: Please Change the table numbering in the supplemental material.  Table S3 is currently the raw transcriptome statistics data.

Response: The table numbering in the supplemental material have been changed (see line 423).

20) Line 431: 2−ΔΔCT method should be referenced. Also, the sample used as a reference to calculate the relative expression data should be indicated.

Response: 2−ΔΔCT method have been referenced, the 18S used as a reference to calculate the relative expression data have been indicated (see line 430-432, 429).

21) Lines 467-468: Metabolome data accession in MetaboLights database is still absent.

Response: The metabolome data has been uploaded and is awaiting approval from the website.

22) The resolution of figures 2, 3, 4, 5, and 6 is still of poor quality in the PDF version of the manuscript. Please be advised.

Response: Dear experts, I have uploaded a clear version of all the figures separately in the magazine for reference.

23) Figure S2 Legend: “Gene Ontology enrichment analysis of differentially expressed genes during development of Pinus koraiensis seed kernels.”

Response: According to the opinion and reminder of the reviewer, the sentences have been rephrased in the Figure S2 legend (see line 5-7).

24) Figure S3: RT-qPCR data needs statistical analysis. Significant changes in gene expression should be indicated in the graphs. 

Response: According to the opinion and reminder of the reviewer, we have added statistical analysis (see Figure S3).

25) Figure S3 legend: S1-S5 stages should be defined in figure legend. The sample used as a reference to calculate the relative expression data should also be indicated.

Response: According to the opinion and reminder of the reviewer, S1-S5 stages have been defined in figure legend. And the 18S used as a reference to calculate the relative expression data have been indicated

Round 3

Reviewer 1 Report

The authors have corrected most of the issues. However, there are still a couple of minor points that need to be addressed to ensure manuscript’s suitability for publication.

a) Line 450:  It is possible that there was a misunderstanding regarding my concerns about the description of the RT-qPCR methodology. Apart from using a reference gene (a gene consistently expressed across all samples in your study, such as 18S), the delta-delta Ct method requires a reference sample (also known as a calibrator or control sample) to calculate the delta-delta Ct values. These values are then used to calculate the relative expression values (2-delta-delta Ct). Consequently, it is important to include information about which specific sample (S1, S2, S3, S4, or S5) was used as a reference both in the Materials and Methods section and in the legend of Figure S3

b) Figure S3 legend: Details of statistical analysis of RT-qPCR data (methodology, meaning of the letters) should be included in figure legend. Also, the information of the sample (S1, S2, S3 S4 or S5) used as a reference to calculate the relative expression data should be included.

c) Figure 1f: Looking the size of error bars, I am surprised that differences in seed weight of samples S1, S2 and S3 appear as statistically significant. Please double check them.

Author Response

List of responses

Manuscript ID: ijms-2480670

Widely targeted metabolomic profiling combined with transcriptome analysis provides new insights into lipid biosynthe-sis in seed kernels of Pinus koraiensis

Dear Editor-in-chief and Reviewers:

Thank you for your letter and for the reviewers’ comments concerning our manuscript entitled “Widely targeted metabolomic profiling combined with transcriptome analysis provides new insights into lipid biosynthe-sis in seed kernels of Pinus koraiensis” (ijms-2480670). Those comments are all valuable and very helpful for revising and improving the quality our paper. We have studied all comments carefully and have made corrections which we hope meet the standard of your highly esteemed journal. For ease of tracking, we highlighted the changes with red color in the revised version. The main corrections in the paper and the responses to the reviewers’ comments are given below.

Responds to reviewer’s comments:

Reviewer #1

The authors have corrected most of the issues. However, there are still a couple of minor points that need to be addressed to ensure manuscript’s suitability for publication.

1) Line 450:  It is possible that there was a misunderstanding regarding my concerns about the description of the RT-qPCR methodology. Apart from using a reference gene (a gene consistently expressed across all samples in your study, such as 18S), the delta-delta Ct method requires a reference sample (also known as a calibrator or control sample) to calculate the delta-delta Ct values. These values are then used to calculate the relative expression values (2-delta-delta Ct). Consequently, it is important to include information about which specific sample (S1, S2, S3, S4, or S5) was used as a reference both in the Materials and Methods section and in the legend of Figure S3

Response: Thanks for the reviewer's professional advice. In this study, S1 stage was used as a reference sample to calculate the 2-delta-delta Ct value. This content have been added (see line 432-433).

2) Figure S3 legend: Details of statistical analysis of RT-qPCR data (methodology, meaning of the letters) should be included in figure legend. Also, the information of the sample (S1, S2, S3 S4 or S5) used as a reference to calculate the relative expression data should be included.

Response: According to the opinion and reminder of the reviewer, the content have been added to Figure S3 legend (see Figure S3).

3) Figure 1f: Looking the size of error bars, I am surprised that differences in seed weight of samples S1, S2 and S3 appear as statistically significant. Please double check them.

Response: Thanks a lot. According to the reminder of the reviewer, we deleted the extremes in the data and performed a multiple comparison analysis again. Thank you again for your opinion.
